# TNF Receptor Associated Factor 2 (TRAF2) Signaling in Cancer

**DOI:** 10.3390/cancers14164055

**Published:** 2022-08-22

**Authors:** Daniela Siegmund, Jennifer Wagner, Harald Wajant

**Affiliations:** Division of Molecular Internal Medicine, Department of Internal Medicine II, University Hospital Würzburg, 97080 Würzburg, Germany

**Keywords:** apoptosis, autophagy, B-cell lymphoma, cellular inhibitor of apoptosis 1/2 (cIAP1/2), necroptosis, nuclear factor ‘kappa-light-chain-enhancer’ of activated B-cells (NFκB), tumor necrosis factor (TNF), TNF receptor associated factor 2 (TRAF2)

## Abstract

**Simple Summary:**

Tumor necrosis factor (TNF) receptor associated factor-2 (TRAF2) is an intracellular adapter protein with E3 ligase activity, which interacts with a plethora of other signaling proteins, including plasma membrane receptors, kinases, phosphatases, other E3 ligases, and deubiquitinases. TRAF2 is involved in various cancer-relevant cellular processes, such as the activation of transcription factors of the NFκB family, stimulation of mitogen-activated protein (MAP) kinase cascades, endoplasmic reticulum (ER) stress signaling, autophagy, and the control of cell death programs. In a context-dependent manner, TRAF2 promotes tumor development but it can also act as a tumor suppressor. Based on a general description, how TRAF2 in concert with TRAF2-interacting proteins and other TRAF proteins act at the molecular level is discussed for its importance for tumor development and its potential usefulness as a therapeutic target in cancer therapy.

**Abstract:**

Tumor necrosis factor (TNF) receptor associated factor-2 (TRAF2) has been originally identified as a protein interacting with TNF receptor 2 (TNFR2) but also binds to several other receptors of the TNF receptor superfamily (TNFRSF). TRAF2, often in concert with other members of the TRAF protein family, is involved in the activation of the classical NFκB pathway and the stimulation of various mitogen-activated protein (MAP) kinase cascades by TNFRSF receptors (TNFRs), but is also required to inhibit the alternative NFκB pathway. TRAF2 has also been implicated in endoplasmic reticulum (ER) stress signaling, the regulation of autophagy, and the control of cell death programs. TRAF2 fulfills its functions by acting as a scaffold, bringing together the E3 ligase cellular inhibitor of apoptosis-1 (cIAP1) and cIAP2 with their substrates and various regulatory proteins, e.g., deubiquitinases. Furthermore, TRAF2 can act as an E3 ligase by help of its N-terminal really interesting new gene (RING) domain. The finding that TRAF2 (but also several other members of the TRAF family) interacts with the latent membrane protein 1 (LMP1) oncogene of the Epstein–Barr virus (EBV) indicated early on that TRAF2 could play a role in the oncogenesis of B-cell malignancies and EBV-associated non-keratinizing nasopharyngeal carcinoma (NPC). TRAF2 can also act as an oncogene in solid tumors, e.g., in colon cancer by promoting Wnt/β-catenin signaling. Moreover, tumor cell-expressed TRAF2 has been identified as a major factor-limiting cancer cell killing by cytotoxic T-cells after immune checkpoint blockade. However, TRAF2 can also be context-dependent as a tumor suppressor, presumably by virtue of its inhibitory effect on the alternative NFκB pathway. For example, inactivating mutations of TRAF2 have been associated with tumor development, e.g., in multiple myeloma and mantle cell lymphoma. In this review, we summarize the various TRAF2-related signaling pathways and their relevance for the oncogenic and tumor suppressive activities of TRAF2. Particularly, we discuss currently emerging concepts to target TRAF2 for therapeutic purposes.

## 1. Introduction

In pioneering work in the mid-1990s, the group of D. Goeddel identified four proteins recruiting to tumor necrosis factor (TNF) receptor 2 (TNFR2). Two of these proteins indicated homology to the just previously identified baculovirus-encoded inhibitor of apoptosis proteins and were accordingly named cellular inhibitor of apoptosis 1 (cIAP1) and -2 (cIAP2) [1]. The two other proteins demonstrated no homologies to proteins known at that time, but shared a conserved C-terminal stretch of app. 200 aa. The two proteins have been named TNF receptor-associated factor-1 (TRAF1) and -2 (TRAF2) and the C-terminal homology domain accordingly as TRAF domain [2]. A C-terminal TRAF domain has also been discovered in four other human proteins, named TRAF3 to TRAF6, and has been subdivided in the compact coiled-coil TRAF-N domain mediating trimerization, and the more loosely packed TRAF-C domain, which in the case of TRAF2 mediates binding to a short aa motif in the cytoplasmic domain of TNFR2 (Figure 1). With exception of TRAF1, the TRAF proteins also share a common N-terminal domain architecture composed of an interesting new gene (RING) domain followed by 5-7 zinc fingers. While activated TNFR2 directly binds TRAF2 and TRAF1, cIAP1 and cIAP2 are indirectly recruited to TNFR2 by help of TRAF2. In fact, eventually TRAF2 seems to fulfill many of its functions in concert with these proteins.

Early after its identification as part of the TNFR2 signaling complex, it has been recognized that TRAF2 is also recruited to the majority of other receptors of the TNF receptor superfamily (TNFRSF) including nearly all TNFRSF receptors (TNFRs) with a TRAF binding motif (direct TRAF2 binding, see Table 1) and all TNFRs with a death domain (DD) (indirect TRAF2 binding). Later, it also became evident that TRAF2 acts beyond the TNFRSF in the signal transduction by other immune regulatory receptors, including members of the toll-like receptor (TLR) family, the type I interferon receptor and the retinoic acid-inducible gene I (RIG I)-like receptor (RLR) family of intracellular pattern recognition receptors recognizing viral RNA (Table 1). Finally, yet importantly, TRAF2 has been implicated in autophagy and endoplasmic reticulum (ER) stress signaling. TRAF2 fulfills its functions primarily by acting as a scaffold, which in a signal-inducible or constitutive manner brings together E3 ligases, their substrates, and also a variety of regulatory factors, including deubiquitinating enzymes (Table 1). 

The most important TRAF2-interacting E3 ligases are cIAP1 and cIAP2. Prominent substrates of the TRAF2-cIAP1/2 complex are kinases, such as transforming growth factor-β (TGF-β)-activated kinase 1 (TAK1), NFκB-inducing kinase (NIK), apoptosis signal-regulating kinase 1 (ASK1), and receptor-interacting kinase 1(RIPK1), which regulate NFκB signaling and induction of programmed cell death. Intriguingly, in some special cases TRAF2 may also act itself as an E3 ligase by help of its RING domain, such as in context of TRAIL death receptor signaling where TRAF2 K48-ubiquitinates caspase-8 [114,115]. Like most other TRAF family members, TRAF2 is involved in the engagement of signaling pathways resulting in the activation of transcription factors, such as the two NFκB pathways, various mitogen-activated protein (MAP) kinase cascades, and the MAVS/TBK1/IRF3 pathway. However, TRAF2 can also affect cellular functions independent from transcription-stimulating pathways by triggering phosphorylation and/or ubiquitination of proteins, thereby regulating their activity, stability, function, or localization. Examples therefore are K48-ubiquitiantion and proteasomal degradation of caspase-8, cRel, interferon regulatory factor 5 (IRF5), and unc-51-like autophagy activating kinase 1 (ULK1) triggered alone by TRAF2 (caspase-8) or by TRAF2 in concert with TRAF3 and cIAP1 and cIAP2 (cRel, IRF5, ULK1) [114,115,116,117]. Further examples of “transcription”-independent TRAF2 activities are the engagement of the Src homology 3 domain-containing guanine nucleotide exchange factor (SGEF), leading to glioblastoma cell migration in response to Fn14 activation [94], and K63-ubiquitination of dual-specificity tyrosine phosphorylation-regulated kinase 1A (DYRK1A), promoting its translocation to vesicles to attenuate epidermal growth factor receptor (EGFR) degradation [51] and its role in mitophagy [85,118,119].

## 2. Role of TRAF2 in Immune Signaling Pathways 

### 2.1. TRAF2 and Activation of the Classical NFκB Pathway 

Nuclear factor kappa-light-chain-enhancer of activated B-cells (NFκB) are homo- and heterodimeric transcription factors formed of the five NFκB proteins p65/RelA, RelB, cRel, p50, and p52, of which the latter two are initially expressed in the form of precursor molecules (p100 and p105). NFκB dimers are held in check by cytoplasmic retention [120,121] resulting from the fact that the nuclear localization sequence (NLS) of NFκBs is blocked in non-stimulated cells by either of two related mechanisms. First, by forming a ternary complex with ankyrin-repeat containing inhibitor of κB proteins (IκBs), e.g., IκBα, or second, by incomplete maturation of the precursor proteins p100 and p105 containing a C-terminal autoinhibitory ankyrin-repeat domain. There are two distinct signaling mechanisms that relieve the NLS of NFκBs from the inhibitory interaction with ankyrin repeats: firstly, the IκB kinase (IKK) complex-induced degradation of IκB proteins and the IKK-induced processing of p105 (classcial or canonical NFκB pathway) and secondly, the NIK-induced processing of p100 to p52 (alternative or non-canonical NFκB pathway) (Figure 2).

A major function of TRAF2 is to transduce activating signals from cell surface receptors, particularly TNFRs, to the IKK complex in the classical NFκB pathway. The latter phosphorylates IκBα and related IκBs to trigger their proteasomal degradation, the key event in activation of the classical NFκB pathway (Figure 2). To fulfill its tasks in classical NFκB signaling, TRAF2 directly or indirectly recruits to the liganded receptor molecules along with the TRAF2-interacting E3 ligases cIAP1 and cIAP2. This results in the activation of the latter. The cIAPs in turn K63 ubiquitinate TRAF2 and other proteins present in the receptor signaling complexes, and thereby create docking sites facilitating the recruitment of the linear ubiquitin assembly complex (LUBAC). The latter catalyzes the formation of linear M1-linked ubiquitin chains, creating binding sites for the NFκB essential modulator (NEMO), a subunit of the IKK complex, and the TAK1-binding-protein-2 (TAB2) subunit of the IKK-engaging TAB2-TAK1 complex. Worth mentioning, TRAF2 also triggers the recruitment of regulatory proteins, such as deubiquitinases, that terminate/resolve the ubiquitination events leading to IKK activation. For example, the cylindromatosis tumor suppressor (Cyld) directly interacts with TRAF2 and removes K63-linked polyubiquitin chains from TRAF2, resulting in reduced NFκB signaling and enhanced apoptosis but also in the maintenance of hematopoietic stem cell dormancy by inhibition of p38 MAP kinase signaling [49,122,123,124]. Likewise, A20 [also named TNFα-induced protein 3 (TNFAIP3)] acts as a K63 deubiquitinase, e.g., for RIPK1, NEMO/IKKγ, or caspase-8, but also, alone or in concert with other E3 ligases, as a K48 E3 ligase [125]. In accordance with the function of its major substrates and in view of the fact that A20 itself is a NFκB target, A20 has been implicated in the downregulation of the classical NFκB pathway and the control of cell death [125].

The generalized mechanisms of receptor-induced TRAF2-mediated activation of the IKK complex described above have been primarily investigated for TNFR1, but there is evidence that similar or related mechanisms also apply for other receptors. For example, cIAPs and/or the LUBAC have also been implicated in NFκB activation by other TRAF2-utilizing receptors, such as CD40, TNFR2, Fn14, and the TRAIL death receptors [126,127,128,129]. Intriguingly, the NFκB-inhibitory effects of dominant-negative TRAF2 mutants on TNFR signaling reported in early years is often more pronounced than the inhibitory effect observed in receptor stimulated TRAF2-deficient cells. A possible explanation for this is that other TRAF proteins, which use overlapping binding sites to TRAF2 in the considered TNFR type, act redundantly with TRAF2 and/or fulfill functions distinct of those of TRAF2. In fact, there is evidence that TRAF2 and TRAF5 act redundantly in TNF-induced classical NFκB signaling and that TRAF2, in cooperation with TRAF1 and TRAF6, redundantly signal CD40-induced NFκB activation [130,131]. Furthermore, it is well-established that TRAF2 cooperates with TRAF3 in the control of alternative NFκB signaling (see 2.2.). In general, however, redundancy and/or cooperativity between TRAF2 and other TRAF proteins have been limitedly investigated so far.

### 2.2. TRAF2 and Activation of the Alternative NFκB Pathway 

TRAF2 and the cIAPs play a central role in the control of the alternative NFκB signaling pathway. In the cytoplasm TRAF2 interacts via TRAF3 with NIK which is constitutively active and enables cIAP1/2-mediated K48-ubiquitination of the latter, resulting in its proteasomal degradation [132,133]. NIK activates IKKα which in turn phosphorylates p100, triggering its proteasomal processing to p52. Therefore, TRAF2, TRAF3, and the cIAPs finally inhibit the alternative NFκB pathway. Thus, in the classical NFκB pathway, TRAF2 and the cIAPs trigger the degradation of pathway inhibitory ankyrin-repeat proteins or ankyrin-repeat domains (IκBs, ankyrin domain of p105), whereas in the alternative NFκB pathway, the same proteins prevent, together with TRAF3, the degradation of a pathway inhibitory ankyrin-repeat domain (Figure 2). The alternative NFκB pathway is typically engaged by members of the TNFRSF, such as Fn14, CD40, TNFR2, and the LTβR. In view of the opposing effects of TRAF2 and the cIAPs on ankyrin-repeat containing NFκB-inhibitory proteins/protein domains in the two NFκB signaling pathways, it first seems counter-intuitive that activation of TNFRs results in the concomitant activation of both pathways. However, this apparent contradiction is resolved when two points are considered: i) that the amount of cell-expressed TRAF2 and cIAP1/2 molecules is limited and ii) that TRAF2 and the cIAPs, along with TRAF3, act constitutively in the cytoplasm of unstimulated cells in context of the alternative NFκB pathway but fulfill their role in the classical NFκB pathway in an inducible manner in plasma membrane-associated receptor signaling complexes. Ligand-induced recruitment of TRAF2 (and/or TRAF3) and the cIAPs to plasma membrane-receptors is accordingly intimately linked to the depletion of these molecules from the cytosol, resulting not only in the formation of classical NFκB-stimulating receptor complexes but also in a reduction of the cytosolic available amount of TRAF2-cIAP1/2 complexes that can be recruited via TRAF3 to NIK to inhibit the alternative NFκB pathway. It is worth mentioning that the sole depletion of TRAF2, TRAF3, and the cIAPs from the cytoplasm is sufficient to engage the alternative NFκB pathway [132,133,134,135] but that this mechanism can be enhanced in its effects by receptor-associated degradation of the TRAFs and the cIAPs. Taken together, despite the opposing quality of the activity of TRAF2 and the cIAPs on the two NFκB signaling pathways, ligand-induced receptor-TRAF2 interaction eventually results in concomitant activation of both pathways.

### 2.3. TRAF2 in RLR Signaling

In RNA virus-infected cells cytosolic double-stranded (ds) RNA is recognized by RIG1 and/or the RIG1-like receptor (RLR) melanoma differentiation-associated 5 (MDA5) and laboratory of genetics and physiology 2 (LGP2) [136]. Binding of dsRNA by RIG1 and MDA5 enables these proteins to convert from an autoinhibited form to a tetrameric “open form” which, assisted by the E3 ligase RIPLET and K63-polyubiquitination, assembles into filaments [137,138,139] (Figure 3). 

The RLR filaments in turn bind to mitochondria antiviral signaling protein (MAVS; also named VISA, IPS-1, or Cardif) and nucleate the formation MAVS filaments [140]. The latter in turn act as signaling platforms, like aggregated TNFRs, to recruit TRAF2, TRAF3, TRAF5, and TRAF6 along with IRF3 and the TRAF-interacting IKK- and TBK1/IKKε complexes to engage downstream signaling pathways, namely the classical NFκB pathway and the TBK1/IRF3/IFNβ pathway [74,141,142]. TRAF2 and TRAF5 on the one side and TRAF3 and TRAF6 on the other side bind to different binding motifs in MAVS and act redundantly to activate a strong innate immune response [141]. While the RING domains of TRAF2 and the other TRAFs were found to be important to mediate NEMO ubiquitination and IKK activation in context of RLR signaling, they appeared dispensable for activation of the TBK1-IRF3 axis [74]. In contrast to receptors of the TNFRSF, RIG1 and MDA5 not only stimulate the activation of NFκB transcription factors by help of the TRAF proteins but also engage IRF3 and IRF7. The reasons for the different signaling qualities of TRAF2 and the other TRAFs in TNFR versus RLR signaling are still unclear. It is worth mentioning that the RLR LGP2 associates with the C-terminus of TRAF2, TRAF3, TRAF5, and TRAF6 and acts as a pan-inhibitor of stimuli using these TRAF proteins for activation of the classical NFκB pathway [70].

## 3. TRAF2 in the Control and Integration of Cell Death Programs, ER Stress and Autophagy 

### 3.1. TRAF2 and Programmed Cell Death 

Soon after it was discovered that TRAF2 not only interacts with TNFR2 but is also recruited to the death domain (DD)-containing TNFR TNFR1 by virtue of the death domain-containing adapter protein TRADD [106,143], it became evident that TRAF2 restricts the ability of TNFR1 to induce caspase-8 activation and apoptosis [144,145,146]. Notably, TNFR2 and other TRAF2-interacting TNFRs, particularly Fn14, not only engage the alternative NFκB pathway but also sensitize for TNFR1-induced cell death by lowering the TNFR1-accessible pool of TRAF2 and cIAPs [134,135,145,147,148,149,150,151,152]. Intriguingly, TRAF2 is not only recruited to the TNFR1 signaling complex within seconds to few minutes but can also be part of TNFR1-induced delayed formed cytosolic complexes enabling caspase-8 activation and apoptosis (complex IIa) or RIPK1 phosphorylation and necroptosis (complex IIb). Inhibition or depletion of cIAPs largely mirrors the effects of TRAF2 depletion on TNFR1 signaling. Therefore, it is tempting to speculate that a significant part of the effects of TRAF2 on TNFR1 signaling is based on its ability to recruit the cIAPs to the TNFR1 signaling complex and to the cytosolic caspase-8/RIPK1-containing complexes IIa and IIb. While TRAF2 molecules associated with TNFR1 seem to be sufficient to fulfill the function of TRAF2 as a transducer of TNFR1-induced classical NFκB signaling, TRAF2 associated with complex IIa and IIb appears to act as an inhibitor of caspase-8 and RIPK1 kinase activation. The NFκB-stimulating activity of TNFR1 signaling complex-associated TRAF2 has been attributed to the K63-ubiquitination of several TNFR1 signaling complex proteins, particularly RIPK1, by the TRAF2-associated cIAPs enabling efficient recruitment of the LUBAC, the TAB2-TAK1 complex and the IKK complex (see also Section 2.1. above). 

Initially, the antiapoptotic activity of TRAF2 was attributed to its relevance for activation of the classical NFκB pathway which results in upregulation of various antiapoptotic proteins, including cFLIP, cIAP2, B-cell lymphoma 2 (Bcl-2) and Bcl-xL, and many more [144,145,153,154,155]. However, a key observation suggested that TRAF2 has antiapoptotic activity, at least in TNFR1 signaling, independent from induction of NFκB-regulated antiapoptotic factors. When cells were primed for a few hours via TNFR2, TRAF2-associated NFκB signaling-promoting functions in the TNFR1 signaling complex, such as RIPK1 ubiquitination and IKK recruitment, were severely affected [151]. In contrast, when TNFR1 and TNFR2 are costimulated, TNFR2-induced depletion of TRAF2 which requires 1-3 h to become apparently sensitized for TNFR1-induced apoptosis occurring with similar slow kinetics but fails now to inhibit the activity and assembly of the rapidly formed classical NFκB-stimulating TNFR1 signaling complex [134,151]. How TRAF2 inhibits TNFR1-induced caspase-8 maturation, which takes place in complex IIa in the latter NFκB-independent scenario, is not fully understood. A part of the explanation could be that TRAF2 K48-ubiquitinates caspase-8 to promote proteasomal degradation of the p43 and p18 fragments of caspase-8. These fragments are generated during death receptor-induced processing of procaspase-8 to mature heterotetrameric caspase-8, and their TRAF2-mediated degradation has been described in context of TRAIL death receptor signaling as a way to downregulate TRAIL sensitivity [114,115]. Worth mentioning, the E3 ligases cIAP1 and cIAP2 are dispensable for TRAF2-mediated ubiquitination of caspase-8 and instead it seems that TRAF2 itself acts here as an E3 ligase by virtue of its RING domain [114]. Death receptor-induced caspase-8 activation normally inhibits DR-induced necroptosis by cleavage of RIPK1 (and possibly other caspase-8 substrates) so that there is no induction of necroptotic cell death [156]. However, when caspase-8 is inhibited, e.g., by drugs or pathogen-encoded proteins, this strongly proinflammatory form of programmed cell death occurs. Interestingly, although TRAF2 acts as an inhibitor of the necroptosis inhibitor caspase-8 as just described, it has been discovered that TRAF2 also protects from DR-induced necroptosis [157,158]. Thus, it appears that the antinecroptotic activity of TRAF2 in DR signaling overrides its potential pronecroptotic activity resulting from caspase-8 inhibition. The antinecroptotic activity has been again traced back to K63 ubiquitination of RIPK1, however, yet in context of complex IIa and/or the RIPK1-RIPK3 necrosome. Sequestration of TRAF2 to liganded TRAF-interacting TNFRs, such as Fn14, consequently results in enhancement of death receptor- and TLR3-induced necroptosis, too [157,159]. The in vivo relevance of the cell death-sensitizing activity of TRAF2/cIAP sequestration by TRAF2-interacting TNFRs is poorly understood. Animal studies suggest that this mechanism is responsible for the high sensitivity of intestinal epithelial cells for TNF-induced killing but could also be operative in the cell death occurring after acute kidney injury or to pathogen-induced hyperinflammation in patients suffering on X-linked inhibitor of apoptosis (XIAP) deficiency [160,161,162,163]. Antiapoptotic activities of TRAF2 have also been described in scenarios beyond death receptor signaling, for example in UV- and oxidative stress-induced apoptosis [155,164]. 

Apoptosis induction by death receptors but also by drugs or other stressors of cellular homeostasis (see also below Section 3.2) can imply generation of reactive oxygen species (ROS) and sustained activation of the cJun N-terminal kinase (JNK) pathway as an enhancing or even essential mechanism [165]. Many of the triggers of apoptotic JNK signaling engage the JNK pathway by TRAF2-dependent activation of the MAP3K apoptosis signal-regulating kinase 1 (ASK1) (e.g., [166,167,168,169]). The latter is kept inactive in cells by binding to thioredoxin (Trx). ROS formation, e.g., in response to TNF, results in dissociation of the ASK1-Trx complex giving TRAF2 access to ASK1 [170]. The central role of the TRAF2-ASK1 interaction for ASK1-driven JNK signaling is also reflected by the fact that the TRAF2-ASK1 interaction is controlled/modulated by various proteins, such as ASK1-interacting protein 1 (AIP1), which enhances TRAF2-ASK1 complex formation, calcium and integrin-binding protein 1 (CIB1), which interferes with TRAF2-ASK1 interaction [171] and the protein arginine methyltransferases-1 (PRMT1), which modifies ASK1 and stabilizes its association with Trx [172]. Worth mentioning, TRAF2 has also been found to bind and stimulate the mammalian ste20-like kinase 1 (MST1) which is also activated by reactive oxygen species and stimulate apoptotic JNK signaling [79,173]. Moreover, TRAF2-MST1 interaction and MST1 activation occur downstream of the ROS-induced dissociation of an inactive Trx-MST1 complex [79]. However, whether and how TRAF2, MST1, and ASK1 act together is currently unclear. 

### 3.2. TRAF2 in ER Stress and Autophagy

The unfolded protein response (UPR) occurs in reaction to ER stress due to accumulation of misfolded proteins. If ER stress is moderate the UPR protect cells by triggering the production of factors (e.g., chaperones) helping to restore fidelity of protein folding, to maintain general functionality and survival. In situations of chronic and strong ER stress, however, the UPR can also trigger cell death programs [174,175]. There are three major sensor proteins for ER stress: (i) activating transcription factor-6 (ATF6), a ER residing transmembrane protein with a cytosolic transcription factor domain which can be released by proteolytic processing, (ii) protein kinase RNA-like ER kinase (PERK), which reduces general protein synthesis by phosphorylation of eukaryotic initiation factor 2alpha (eIF2α) but also promotes enhanced translation of selected mRNAs including that encoding the transcription factor ATF4 and (iii) the bifunctional inositol-requiring enzyme 1alpha [IRE1α, also named ER to nucleus signaling 1 (ERN1)] which has serine/threonine-protein kinase and RNAse activity. Upon sensing ER stress by yet poorly understood mechanisms (dissociation from the chaperone GRP-78/BiP and/or direct binding of misfolded proteins), IRE1α dimerizes and becomes activated by trans-autophosphorylation. This stimulates the RNAse activity of IRE1α and enables the recruitment of TRAF2 [174,176] (Figure 4). With the help of its RNAse activity, IRE1α splices out in a spliceosome-independent manner a small intron from the mRNA of the transcription factor XBP1, eventually resulting in a switch from the production of an inactive “unspliced” form (XBP1u) to an active transcription factor (XBP1s) but also cleaves various RNAs resulting in IRE1-dependent decay of mRNA (RIDD) [177]. Interaction of TRAF2 with phosphorylated dimerized IRE1α leads to the recruitment and activation of ASK1 and stimulation of MAP kinase signaling pathways resulting in the engagement of p38, extracellular-regulated kinase (ERK) but particularly of JNKs (Figure 4). However, the IRE1α/TRAF2 dyad can also promote activation of the classical NFκB pathway or cell death programs (Figure 4). 

IRE1α-induced TRAF2-mediated NFκB activation involves K63-ubiquitination of RIPK1 [178]. Furthermore, it has been discovered that the IRE1α/TRAF2/ASK1 complex recruits TRADD and FADD in response to ER stress [179]. The recruitment of these DD-containing proteins resembles the situation in death receptor-signaling but this similarity has yet not evaluated in molecular detail. In an early phase, IRE1α/TRAF2-mediated JNK signaling contributes to cell survival by inducing transcription of cIAP1, cIAP2, and xIAP [180]. Sustained JNK signaling, however, can then become proapoptotic by activation of the proapoptotic Bcl-2 family members BIM and BID and inhibition of antiapoptotic Bcl-2 family members, such as Bcl-2, Bcl-xL and MCL1 [174]. It is worth mentioning that the proapoptotic IRE1α/TRAF2/ASK1/JNK axis might cooperate with the PERK/ATF4-dependent induction of TRAIL death receptors [174,181] and/or the IRE1α/TRAF2/IKK-induced production of TNF [182]. TRAF2-deficient MEFs are more susceptible for ER stress-induced apoptosis than wild-type MEFs [183] but whether this is due to the role of TRAF2 in ER stress/IRE1α signaling or rather reflect defects in other TRAF2-dependent activities affecting the general apoptosis sensitivity of cells is unclear. The IRE1α/TRAF2/ASK1/JNK axis is also connected with autophagy as discussed below.

Autophagy, which occurs in various forms, such as macroautophagy, microautophagy, chaperone-mediated autophagy, or selective autophagy for certain proteins, is a cellular housekeeping activity to purge intracellular “waste”, e.g., protein aggregates, damaged organelles and intracellular pathogens, but is also important for maintenance of plasma membrane integrity [184,185,186,187]. In view of these activities, it is not surprising that steady-state autophagy is increased when cellular homeostasis is disturbed, e.g., by nutrient deficiency, genomic instability or infection, but also by ER stress and cell death programs. Since TRAF2 plays a role in IRE1α-signaling, regulation of cell death and the NFκB system, it can also be of relevance for autophagy. Indeed, TRAF2 and TRAF2-associated proteins have been implicated in several ways in the crosstalk between autophagy, ER stress, cell death and inflammation, particularly in context of cancer development and cancer treatment.

Bcl-2 and Bcl-xL attenuate autophagy by binding and inhibition of Beclin-1, thereby hindering the latter to stimulate the ULK1 complex, which promotes autophagosome initiating vesicle nucleation [188]. Accordingly, BH3-only proteins competing with Beclin for Bcl-2 binding and JNK-mediated phosphorylation of Bcl-2, resulting in reduced Beclin-1/Bcl-2 interaction, which are able to enhance autophagy [189,190]. Therefore, apoptosis inducers engaging these mechanisms can have a stimulatory effect on autophagy that typically results in reduced cell death [189]. For example, apoptosis-reducing TRAF2- and JNK-mediated stimulation of autophagy have been demonstrated for TRAIL and the IRE1α inhibitor protein Bax inhibitor-1 [191,192,193]. TRAF2 cannot only promote Beclin activation due to its role in JNK stimulating signaling pathways but in concert with cIAP1 and cIAP2 and/or sphingosin kinase 1, and also directly by K63-polyubiquitination of Beclin [43,194]. Worth mentioning, JNK can also inhibit context-dependent autophagy flux. For example, nitrobenzoxadiazole derivatives which antagonizes sequestration of TRAF2 by glutathione transferase and, which are under consideration as anticancer drugs, impair clearance of autophagosomes in a JNK-dependent manner [195].

TRAF2 and cIAP1 have also been found to mediate PTEN-induced kinase 1 (PINK1)/Parkin-independent mitophagy in response to the anti-parasitic lactone ivermectin by promoting ubiquitination and fragmentation of mitochondria [119]. Similarly, the anti-inflammatory triterpene celastrol binds to Nur77, resulting in enhanced TRAF2-Nur77 interaction and eventually mitophagy [85]. In contrast, to ivermectin-induced TRAF2-dependent mitophagy and celastrol-induced TRAF2-Nur77-mediated mitophagy seems to involve PINK1 and Parkin [85]. Indeed, it has been discovered i) that TRAF2 interacts with Parkin in mitophagy induced by the mitochondria-depolarizing compound carbonyl cyanide m-chlorophenyl hydrazine and ii) that TRAF2 partly substitutes Parkin as an E3 ligase in this scenario [118]. Therefore, TRAF2 could contribute to mitophagy in different ways.

## 4. TRAF2 and the NFκB System in Oncogenesis 

### 4.1. TRAF2 in EBV-associated Oncogenesis

Soon after TRAF2 and TRAF1 were described the first time, there was evidence that TRAF2 can also play a role in tumor biology. Both molecules, and later also TRAF3, TRAF5, and TRAF6, were recognized as part of the plasma membrane-associated protein complex instructed by the latent infection membrane protein 1 (LMP1) of Epstein-Barr virus (EBV, human γ -herpesvirus 4) [35,36,196]. Several lymphoproliferative diseases, including malignant ones, are associated with EBV infection, such as Burkitt lymphoma and Hodgkin’s lymphoma, but also non-lymphoid malignancies, particularly nasopharyngeal carcinoma (NPC) and gastric cancer [197]. LMP1 is frequently found in EBV-associated cancers and acts as an oncogene by stimulating various signaling pathways, including those resulting in the activation of NFκB transcription factors and MAP kinases such as JNK, p38, and ERK. In accordance with a central role of the NFκB system in EBV/LMP1-dependent oncogenesis, genomic analysis of NPC revealed in LMP1-independent cases somatic aberrations resulting in constitutive NFκB activation [198]. TRAF2 in addition to TRAF1, TRAF3, TRAF5, and TRAF6 recruit to a PXQXT/S motif in a plasma membrane-proximal domain of the cytoplasmic tail of LMP1 [199,200,201]. This domain has been designated as transformation effector site 1 (TES1) or C-terminal activation region 1 (CTAR1) and is needed, together with a more C-terminally located domain called TES2/CTAR2, for transformation of B-lymphocytes [202,203,204]. TES1/CTAR1 and TES2/CTAR2 cooperate in LMP1-induced TRAF protein-mediated activation of NFκB transcription factors, JNK and p38 [130,203,204]). It is worth mentioning that TES2/CTAR2 recruits TRADD and BS69 and that there is initial evidence that these proteins in turn recruit TRAF2 and TRAF6 to the CTAR2, as well [203,204]. Studies in TRAF2-deficient B-cell lines failed to demonstrate a crucial role of TRAF2 in LMP1 signaling [130,205] while there is clear evidence for the relevance of other TRAF proteins, particularly TRAF3 and TRAF6, in this respect [201,205,206]. Likewise, there were no effects of TRAF2-TRAF5 double deficiency on LMP1-induced nuclear translocation of RelA in embryonal fibroblasts [206]. However, in TRAF2-deficient models there is already considerable constitutive p100 processing, thus constitutive activation of the alternative NFκB pathway. It is therefore challenging in these cell models to draw conclusions on the relevance of the LMP1-TRAF2 interaction for LMP1-induced alternative NFκB signaling.

Since LMP1 aggregates in the plasma membrane and recruits TRAF molecules, it obviously mimics in several, but not all, aspects the ligand-induced signaling complexes of receptors of the TNFRSF. Indeed, activation of both NFκB signaling pathways and JNKs in response to EBV infection has been attributed to LMP1 signaling via TRAF2 and TRAF1 (but also TRAF6), worth mentioning without the involvement cIAP1 and cIAP2 [199,207,208,209]. LMP1-induced NFκB-mediated upregulation of A20 and TRAF1 and subsequent recruitment of these factors along with the LUBAC is another aspect where LMP1 signaling resembles that of TRAF2-interacting TNFRs [36,209,210,211]. Indeed, a genome-wide siRNA screen for proteins involved in LMP1-induced NFκB signaling resulted in the identification of 155 proteins, of which 79 was similarly relevant for TNF-induced NFκB activation [212]. With respect to its activities in B-cells and its ability to recruit TRAF1, TRAF2, TRAF3, TRAF5, and TRAF6, LMP1 largely mimics activated CD40. The general similarities between the signaling complexes of LMP1 and TNFRs, particularly CD40, however, should not hide the fact that there are clear differences. For example, as mentioned above, LMP1 signaling was found to be normal in TRAF2-deficient B-cell lines while CD40 signaling was reduced in the same cell lines and in B-cells of mice with TRAF2-deficient B-cells [130,205,213]. Vice versa, CD40 singling was unaffected in TRAF3-deficient B-cell lines while LMP1 signaling was strongly attenuated [130,205]. Furthermore, CD40, but not a CD40 chimera with the C-terminal cytosolic tail of LMP1, triggers TRAF2-dependent degradation of TRAF3 [214,215]. These differences have been traced back to the higher efficiency with which CD40 recruits TRAF2 [216] and illustrate that not only the ensemble of TRAF proteins interacting with a certain receptor determines the signaling output but also the absolute and relative strengths of the receptor-TRAF protein interactions. 

LMP1 is not the only EBV protein interacting with TRAF2. The EBV protein Na, which is crucially involved in the reactivation of EBV in latent infected cells leading to cell lysis, also interacts with TRAF2 [217,218]. Na utilizes TRAF2 to engage JNK-dependent expression of lytic gene expression [218]. It is worth mentioning that Na overexpression, similar to LMP1, also stimulates alternative but not classical NFκB signaling [218]. This again emphasizes that TRAF2-dependent processes, here TRAF2-Na or LMP1-Na interaction and TRAF2-dependent NIK degradation, might compete for TRAF2 resulting in hardly predictable activities of TRAF2-dependent events. 

### 4.2. TRAF2 and the Alternative NFκB Pathway in Multiple Myeloma and B-cell Lymphoma 

Activation of the NFκB system has been recognized in a variety of tumor types, however, whether this correlates with genetic mutations was poorly investigated for a long time. Gene expression profiling data and functional studies revealed that development of multiple myeloma (MM) is frequently associated with mutations resulting in enhanced NFκB signaling particularly via the alternative NFκB pathway [219,220,221,222]. Most of these mutations are found in the gene of TRAF3 but mutations negatively affecting expression or function of TRAF2 were also frequently found defining TRAF2 (and TRAF3) in MM as a tumor suppressor. Mouse models with deletion of cIAP1 and cIAP2 but not of either protein alone makes the survival of B-cells independent of BaffR signaling and led to their uncontrolled accumulation in vivo, resulting in B-cell lineage malignancies [223]. Thus, as so often the activity of TRAF2 and TRAF3, here as tumor suppressors, require support by cIAP1 and cIAP2. Furthermore, a Crispr/Cas9 genetic screen identified TRAF2 as factor promoting antitumoral IMID activity via activation of alternative NFκB and ERK signaling [224]. Recurrently occurring mutations in TRAF2 were also observed in mantle cell lymphoma resistant against the B-cell receptor (BCR) signaling inhibitor ibrutinib [225].

Constitutive activation of the NFκB system is also a hallmark of diffuse large B-cell lymphoma (DLBCL) particularly of the activated B-cell-like subtype (ABC-DLBCL). In fact, somatic mutations in various genes encoding components of the NFκB system have been identified including mutations in the TRAF2 gene [226,227]. These studies argue for an antitumoral role of TRAF2 in DLBCL, presumably due to its ability to suppress the alternative NFκB pathway. However, there is evidence that TRAF2 can also elicit protumoral activity in DLBCL. Immunohistochemical analyses revealed strong expression of TRAF2 in ABC-like DLBCL with significant association with reduced progression-free survival [228]. Moreover, functional genetic screens in DLBCL cell lines identified TRAF2 as a factor conferring resistance against mucosa-associated lymphoid tissue lymphoma translocation protein 1 (MALT1) inhibitor and the cereblon E3 ligase–modulating agent CC-122 [229,230]. One of the somatic mutations in TRAF2 identified in DLBCL results in enhanced TRAF2-dependent classical NFκB activation [226]. In view of the general relevance of the NFκB system for the physiology and pathophysiology of B-cells, it is tempting to speculate that the contradictory impact of TRAF2 on DLBCL reflects its opposing effects on the classical and alternative NFκB pathway.

In accordance with the notion that the alternative NFκB pathway and therefore also TRAF2 and TRAF3 act as tumor suppressors in B-cells, mice with B-cells deficient in TRAF3 expression demonstrate prolonged B-cell survival and develop spontaneous B-cell lymphoma at a higher age [231,232]. Likewise, transgenic mice expressing Bcl2 and a RING/Zn finger domain-deletion mutant of TRAF2 develop lymphoma with high frequency [233]. Worth mentioning, the TRAF2 deletion mutant expressed in these mice causes degradation of endogenous TRAF2 but also its own degradation, suggesting that it mimics TRAF2 deficiency [234].

### 4.3. TRAF2 and Wnt/β-catenin Signaling 

TRAF2 mutations have also been described for colon cancer [235]. Furthermore, TRAF2 transcript expression is much higher in colorectal cancer compared to benign tissues and is negatively associated with patient survival [46]. A dominant oncogenic driver of colon cancer development is the Wnt/β-catenin signaling pathway culminating in the formation and activation of a nuclear transcriptional complex containing β-catenin, T-cell factor 4 (TCF4) and the TRAF2- and Nck-interacting kinase (TNIK). TRAF2 is not only able to interact with the latter [105] but also binds β-catenin [46]. More important, TRAF2 binds and stabilizes the β-catenin-TNIK-TCF4 complex thereby crucially contributing to TCF4 activation [46]. There is also initial evidence that TRAF2 favor colon cancer development beyond its role in the Wnt/β-catenin. Peng at al [92] could indicate that TRAF2 contributes to EGF-induced ribosomal S6 kinase 2 (RSK2) activation in colon cancer cell lines and Jia et al. [72] reported recently that TRAF2 binds LRPPRC to ubiquitinate argininosuccinate synthase 1 marking this enzyme for degradation resulting in reduced arginine synthesis and the latter is required for tumor growth. Colon cancer is possibly not the only tumor entity where the β-catenin-TRAF2 axis gains relevance. For chronic myelogenous leukemia (CML), it has been reported that the TRAF2-interacting TNFRSF receptor CD27 stimulates the oncogenic Wnt-β-catenin-TNIK-TCF4 pathway, resulting in enhanced proliferation of leukemia stem cells and leukemia progression in a TRAF2-dependent manner [236].

### 4.4. TRAF2 in Breast Cancer and Other Solid Tumors 

The NFκB system in general affects a variety of cancer relevant processes, and it is thus not surprising that TRAF2 via its role in NFκB signaling not only contributes to malignant transformation of lymphoma but also to development of solid cancers. For example, in breast cancer there is ample evidence that NFκB signaling promotes tumor progression, metastasis and resistance against chemo- and radiotherapy ([237,238]). Perhaps most intriguing is the role of the TRAF2-NFκB connection in breast cancer cell transformation by the breast cancer oncogene IκB kinase ε (IKKε; IKKi). IKKε expression is upregulated in >30% of breast cancers and to an even higher frequency in glioblastoma (50%) and pancreatic ductal adenocarcinoma (65%) [239]. IKKε stimulates the K63-ubiquitination activity of the TRAF2-cIAP1 and TRAF2-cIAP2 complexes by phosphorylation of serine 11 on TRAF2 resulting in NFκB activation and various protumoral effects in vitro and in vivo, such as increased cell proliferation, anchorage-independent colony formation, and enhanced tumor formation [66,240]. Protumoral TRAF2 activity in breast carcinoma is also evident from xenogeneic tumor models with human MDA-MB-231 breast cancer cell variants overexpressing TRAF2, which indicated enhanced orthotopic tumor growth in mice after injection into the mammary fat pad and increased skeletal tumors after intra-tibial application [241]. Interestingly, it has been reported that receptor activator of NFκB (RANK)-c, a splice form of the TRAF2-interacting TNFR RANK, which is found in 3.2% of breast cancer patients and which acts in a dominant-negative fashion on RANK-induced NFκB activation, is inversely correlated with disease grade [242,243]. Moreover, RANK-c requires its TRAF2 binding sites to elicit its tumor attenuating NFκB-inhibitory activity pointing to TRAF2 sequestration as mode of action. Furthermore, there is evidence that certain microRNAs (miR-892b, miR-502-5p, miR-205-5p) downregulate TRAF2 expression and thereby suppress breast cancer development [244,245,246]. Similarly, the E3 ligase carboxyl terminus of Hsp70-interacting protein (CHIP), which triggers TRAF2 degradation, has tumor suppressive activity in breast and gastric cancer cells [48,247,248]. Furthermore, a crucial role of the TRAF2-NFκB axis as an adaptive survival mechanism in response to EGFR oncogene inhibition has been reported for lung cancer and there is also evidence that TRAF2 expression confers resistance against irradiation [52,249]. In accordance with the antiapoptotic activities of TRAF2, it has been discovered in various solid cancer cell lines that TRAF2 expression via NFκB activation protects against radio- and chemotherapy but a detailed discussion of this issue is behind the scope of this review. 

## 5. TRAF2 and the Response to Immune Checkpoint Blockade

The approval of immune checkpoint inhibitors (ICIs) brought the clinical breakthrough for immunotherapy. Despite the often-impressive clinical efficacy of immune checkpoint blockade (ICB), however, many patients do not respond or develop dose-limiting autoimmune effects. Therefore, there are considerable ongoing efforts to identify drugs that enhance clinical efficacy of ICB. Two comprehensive independent studies, utilizing different methodologies, argue for TRAF2 as a novel and promising target whose inhibition could synergize with ICIs. Vredevoogd et al. [250] identified in a genome-wide Crispr-Cas9 screen in IFNγ-resistant melanoma cells TRAF2 and cIAP1 as the two major hits increasing the sensitivity for IFNγ-independent killing by CD8^+^ T-cells. In view of the relevance of TRAF2 and the cIAPs to protect from TNFR1-induced cell death (see Section 3.1), this suggested that the sensitivity of tumor cells for TNF cytotoxicity is a crucial factor for ICB efficacy. Aligned with this idea, there were also hits in other factors regulating TNF cytotoxicity. Furthermore, while the mutational status of the TNF system in untreated tumors failed to correlate with patient survival, a positive correlation between TNF expression and the response to anti-PD1 therapy has been observed [250]. Follow-up in vitro and in vivo experiments indeed indicated that TRAF2 deficiency sensitizes tumor cells for killing by TNF expressed by CD8^+^ T cells. Notably, it has been reported in this study, too, that TNF-like weak inducer of apoptosis (TWEAK), the ligand of Fn14, also sensitizes cancer cells for TNF-mediated killing by CD8^+^ T cells. This corresponds very well to the already discussed ability of Fn14 to deplete the cytosolic pool of TRAF2-cIAP1/2 complexes, (see Section 3.1). In a second approach whole-exome and transcriptomic data along with clinical outcome data derived of more than >1000 patients treated with ICIs were evaluated in a meta-analysis for predictors of ICI response. Worth mentioning, copy-number analysis in this study also identified loss of 9q34, the position of the TRAF2 gene, to be positively associated with clinical response [251]. 

## 6. Therapeutic TRAF2 Targeting Strategies 

In view of its often protumoral activities and its ICB antagonizing effects, TRAF2 is an obvious potential target for tumor therapy. However, since TRAF2 utilizes its pleiotropic functions by a variety of binding partners, which interact with different parts of the molecule, it is difficult to define a side in TRAF2 enabling general inhibition of TRAF2. Since some TNFRs deplete cytosolic TRAF2 pools and even trigger its subsequent degradation, agonists of such receptors could be considered as a kind of TRAF2 inhibitor, at least as inhibitor of cytosolic TRAF2 functions. This perception is interesting for targeting the ICB antagonizing effects of TRAF2. Indeed, as discussed above, initial studies indicated that TWEAK acts as an inhibitor of the ICB antagonizing TRAF2 activity [250]. Eventually, agonists of TRAF2-interacting receptors act here as “selective” inhibitors of TRAF2 survival functions (or other cytosolic TRAF2 activities) but not as general TRAF2 inhibitors. Indeed, the TNFRs may even actively exploit TRAF2 to exert proinflammatory activity. Since TRAF2 fulfills many of its activities with essential support of cIAP1 or cIAP2, inhibitors of these molecules can also be considered to be TRAF2 inhibitors. This appears particularly interesting because a variety of cIAP antagonists (also called SMAC mimetics) are under preclinical and clinical evaluation for cancer therapy [252]. The feasibility of the consideration of agonists of cytosolic TRAF2-depleting TNFRs, such as Fn14, and IAP antagonists as pseudo TRAF2 inhibitors, is underscored by the fact that TRAF2 deficiency, Fn14 agonists and IAP antagonists trigger the same cellular hallmarks, namely activation of the alternative NFκB pathway and sensitization for TNF-induced cell death. Nevertheless, TNFR agonists can trigger TRAF2-dependent and independent signaling pathways and IAPs have TRAF2-independent activities, too. Thus, direct TRAF2 inhibitors still have the potential to elicit therapeutic activity beyond TNFR agonists and IAP antagonists.

Liquidambaric acid (LDA, or betulonic acid) is a pentacyclic triterpenoid which has been decades under investigation as an anti-cancer compound [253]. LDA has just been recently identified as a first TRAF2-binding small molecule. LDA prevents β-catenin-TRAF2 interaction and inhibits the Wnt/β-catenin pathway and colon cancer development in a xenogeneic colon cancer model [46]. Interestingly, LDA inhibited the interaction of TRAF2 and β-catenin but demonstrated no effect on TRADD binding of TRAF2 [46]. This example illustrates that it could be possible to develop low molecular weight inhibitors, which specifically antagonize certain aspects of TRAF2 biology.

IAP antagonists demonstrate low toxicity, are well tolerated and displayed good antitumor activity in preclinical models. However, in clinical trials with IAP antagonists as single agents, only limited therapeutic activity has been observed. Because of the intersection of effects of IAP antagonists and TRAF2 deficiency, it appears plausible that TRAF2 inhibitors, similarly to IAP antagonists, may only demonstrate limited activity in patients. It could be therefore necessary to combine such compounds with other drugs to exploit the possible antitumoral potential of TRAF2 inhibition for cancer therapy. Against the background that TRAF2 and cIAPs ([250,254]) protect tumor cells against the cytotoxic action of CD8^+^ T-cells and NK cells after ICB, combination therapies of TRAF2 inhibitors and checkpoint inhibitors appears particularly interesting. Clinical trials with IAP antagonists and ICIs are ongoing [252] and will provide first hints in this direction. 

## 7. Conclusions

A major general conclusion that can been drawn from the available TRAF2 literature is that TRAF2 is part of a large variety of constitutively formed and signal-induced protein complexes that act in quite different cellular processes and in different cellular compartments. Importantly, there is evidence that these TRAF2-dependent processes compete for a limited pool of TRAF2 (and TRAF2-associated factors) resulting in complex context-dependent crosstalk mechanisms. A consequence of increased TRAF2 expression, as often observed in tumor cells, is that competition for TRAF2 loses functional relevance. Therefore, increased TRAF2 expression may not only generally enhance TRAF2-dependent processes but also weaken crosstalk mechanisms that reciprocally counterbalance TRAF2-dependent processes in non-transformed cells. For example, increased TRAF2 expression may not only increase the ability of TRAF2-interacting TNFRs to engage the classical NFκB pathway but will also limit their capacity to sensitize for cell death by TRAF2-cIAP1 sequestration. Thus, deregulated TRAF2 expression could not only change the amplitude of TRAF2-dependent responses but also the qualitative outcome of the whole network of TRAF2-mediated processes. 

A second conclusion is that TRAF2, despite its function as a tumor suppressor in B-cells, is a promising target for tumor therapy, particularly in combination with checkpoint inhibitors. An important and possibly decisive question in this context is whether effective inhibition of TRAF2 can be achieved. Indirect inhibition of TRAF2, e.g., with agonists of TRAF2-depleting receptors, in addition to direct inhibition of TRAF2 by small molecules appears conceivable. Both possibilities have been limitedly investigated so far and define a major challenge in the field in the next years. However, the clinical development of TRAF2 inhibitors should be accompanied by close monitoring for signs of autoimmunity or B-cell hyperplasia.

## Figures and Tables

**Figure 1 cancers-14-04055-f001:**
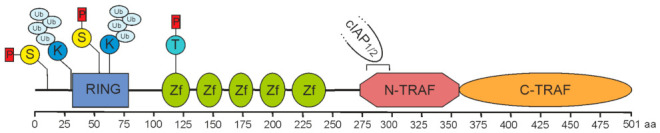
Domain architecture of TRAF2. A sequence of 501 amino acids prescribes the molecular structure of full-length TRAF2. It is essentially composed of a RING domain (aa 32–79), five zinc fingers (Zf) and a TRAF domain comprising a coiled-coil N-TRAF domain with a cIAP1/2 binding site (aa 283–294) [3,4] and a C-TRAF domain. Phosphorylation and ubiquitination sites of known relevance for TRAF2 function are indicated and comprise serine S11 and S55 [5,6,7], lysine K31 and K63 [8,9], and threonine T117 [10].

**Figure 2 cancers-14-04055-f002:**
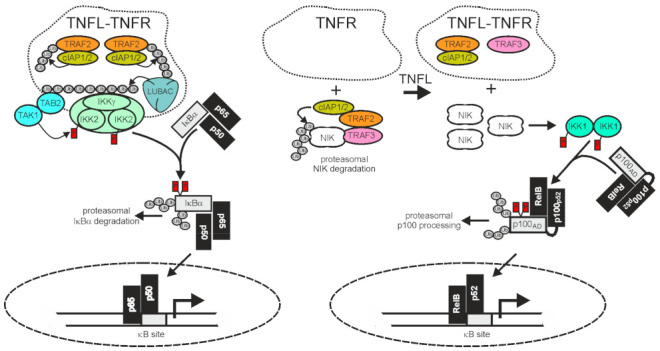
TRAF2 and the cIAPs in receptor-induced activation of the classical and alternative NFκB pathway. The activities of TRAF2 and the cIAPs have opposing qualities in the classical (left panel) and alternative (right panel) NFκB signaling pathway. In the classical NFκB pathway TRAF2 and the cIAPs enable signaling, while in the alternative NFκB pathway they act as inhibitors. Importantly, TNFR-induced recruitment of TRAF2 and the cIAPs, which triggers the classical NFκB pathway, is linked with an “inhibitory” relocation of these molecules away from their cytosolic substrate NIK in the alternative NFκB pathway. Therefore, TNFRs eventually stimulate both NFκB signaling pathways despite the opposing quality they have in these pathways. For more details, refer to main text.

**Figure 3 cancers-14-04055-f003:**
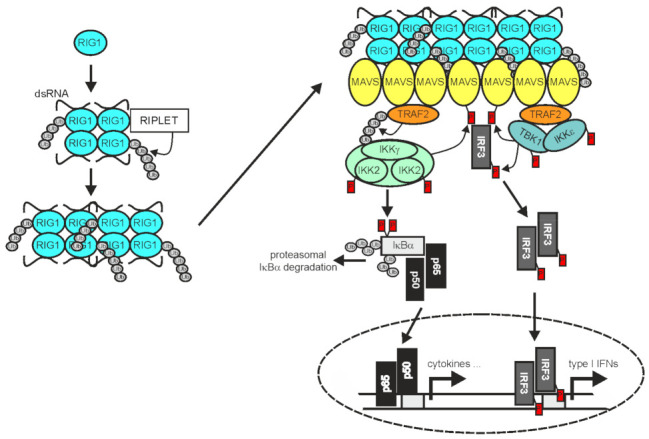
TRAF2 in RIG1 signaling. Binding of dsRNA by RIG1, results in conformational change, K63-ubiquitination, and filament formation. RIG1 filaments in turn instruct filament formation of mitochondria-associated MAVS. The MAVS filaments enable recruitment of TRAF2 and IRF3, but also other TRAF proteins not indicated here. TRAF2 and the other TRAF proteins mediate the recruitment of the IKK complex and TANK-binding kinase 1 (TBK1)/IKKε enabling activation of the classical NFκB pathway and IRF3 by the mechanisms described in detail in the text. Please note, TRAF2 acts independently here from cIAP1 and cIAP2 [74]. For more details refer to main text.

**Figure 4 cancers-14-04055-f004:**
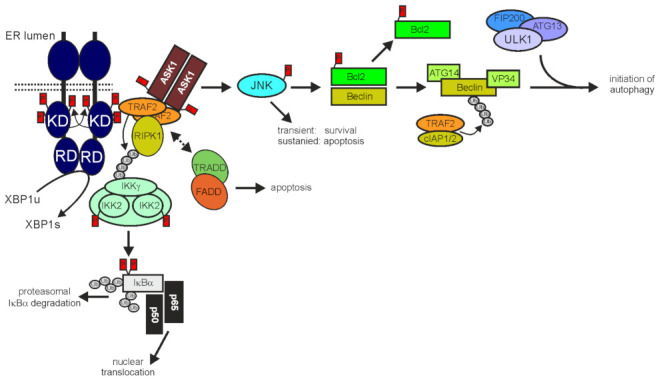
TRAF2 in IER1 signaling. After ER stress-induced dimerization IREα triggers production of the transcription factor XBP1 by its endonuclease activity engages splicing and recruits TRAF2 to trigger ASK1/JNK signaling but also, by still poorly studied mechanisms, classical NFκB signaling and apoptosis. The IRE1α/TRAF2/ASK/JNK axis is also connected with autophagy by the JNK-mediated phosphorylation of the Beclin inhibitory Bcl-2 protein. Interestingly, TRAF2 in concert with cIAP1/2 can further contribute to autophagy by K63-ubiquitination of Beclin. For more details refer to main text.

**Table 1 cancers-14-04055-t001:** TRAF2 interacting proteins.

Protein	Type of Protein	Experimental Evidence	Target Domain in TRAF	Reference
TNFR2	TNFRSF	THS, GST, endo Co-IP	CTD	[2]
LTβR	TNFRSF	endo Co-IP		[11]
OX40	TNFRSF	THS, Co-IP		[12,13]
CD40	TNFRSF	THS, GST, endo Co-IP		[14]
CD27	TNFRSF	THS, Co-IP		[15,16,17]
CD30	TNFRSF	THS, GST		[18,19]
4-1BB	TNFRSF	THS, GST, endo Co-IP		[12,20,21]
RANK	TNFRSF	GST, Co-IP		[22,23,24]
Fn14	TNFRSF	GST		[25]
TACI	TNFRSF	THS, Co-IP		[26]
HVEM	TNFRSF	GST		[27]
NGFR	TNFRSF	Co-IP		[28]
BCMA	TNFRSF	Co-IP		[29]
GITR	TNFRSF	THS, endo Co-IP		[30,31]
TROY	TNFRSF	Co-IP		[32]
IL15R	receptor	Co-IP		[33]
IFNαR1	receptor	GST, Co-IP		[34]
LMP1	viral oncogen	GST, Co-IP		[35,36]
A20	DUB, E3 ligase	THS, Co-IP		[37]
AIP1	Ras-GAP	Co-IP	RING/zinc	[38]
AIMP2	adaptor	THS, GST, endo Co-IP		[39]
APPL1	adaptor	GST, endo Co-IP		[40]
AWP1	adaptor	THS, Co-IP	TD	[41]
Bcl10	adaptor	THS, Co-IP		[42]
Beclin	autophagy	GST, endo Co-IP	RING	[43]
Caspase-2	caspase	Endo Co-IP		[44]
Caspase-12	caspase	Co-IP	NTD	[45]
β-catenin	proto-oncogene	Co-IP, MST	TD	[46]
caveolin-1	plasma membrane protein	endo Co-IP		[47]
CHIP	E3 ligase	endo Co-IP		[48]
cIAP1	E3 ligase	THS, Co-IP	NTD	[1]
cIAP2	E3 ligase	THS, Co-IP	NTD	[1]
CYLD	DUB	THS, Co-IP	TD	[49]
DUSP14	phosphatase	Co-IP		[50]
DYRK1A	kinase	endo Co-IP	TD	[51]
EGFR	kinase	endo Co-IP		[52]
EI24	E3 ligase	Co-IP		[53]
eIF4GI	scaffold	THS, GST, Co-IP	TD	[54]
Eva1	adhesion protein	endo Co-IP		[55]
FAK	kinase	endo Co-IP		[56]
Filamin	actin binder	Co-IP	RZ	[57]
GCKR	kinase	endo Co-IP	TD	[58]
Gpx1	peroxidase	Co-IP	TD	[59]
GRA15	virulence factor	Co-IP		[60]
GSTP1-1	gluthation transferase	Co-IP		[61]
HGK	Kinase	Co-IP		[62]
Hoxa1	transcription factor	THS, Co-IP		[63]
HSP70	Chaperon	Co-IP	TD	[64]
IKK1	kinase	GST, endo Co-IP	RING	[65]
IKK2	kinase	GST, endo Co-IP	RING	[65]
IKKe	kinase	Co-IP		[66]
IRE1	kinase, nuclease	endo Co-IP		[67]
I-TRAF	adaptor	THS, GST, Co-IP	TD	[68]
JIK	kinase	Co-IP		[45]
KRC	DNA binding	endo Co-IP	TD	[69]
LGP2	RLR	Co-IP	CTD	[70]
LILRB3	receptor	endo Co-IP		[71]
LRPPRC	RNA regulation	Co-IP		[72]
MAVS	adaptor	Co-IP	CTD	[73,74]
MEKK1	kinase	Co-IP		[75]
MIZ	transcription factor	GST, endo Co-IP	RING	[76]
MLK3	kinase	Co-IP	NTD	[77,78]
MST1	kinase	endo Co-IP	Zn fingers	[79]
TRIM37	E3 ligase	Co-IP	TD	[80]
Nef	virulence factor	GST		[81,82]
HCV core	virulence factor	GST		[81]
NIP45	transcription factor associated	endo Co-IP		[83]
NSP1	virulence factor	Co-IP		[84]
Nur77	nuclear receptor	Co-IP	RING, NTD	[85]
parkin	E3 ligase	endo Co-IP		[86]
proPTPRN2	phosphatase	Co-IP	RING	[87]
RET/PTC3	oncogenic RTK fusion protein	Co-IP		[88]
RIPK1	kinase	Co-IP	NTD, CTD	[89]
RIP2	kinase	Co-IP		[90]
RNAseT2	ribonuclease	Co-IP		[91]
RSK2	kinase	Co-IP		[92]
SHP-1	phosphatase	Co-IP		[93]
SGEF	GEF	Co-IP	TD	[94]
Sharpin	scaffold	Co-IP		[95]
SIAH-2	E3 ligase	GST		[96]
SMAD4	signaling protein	THS, endo Co-IP		[97]
SMURF-2	E3 ligase	THS, Co-IP		[98]
SMYD2	methyltransferase	Co-IP, SPR		[99]
SphK1	kinase	GST, Co-IP		[100]
T2BP / TIFA	adaptor	THS, Co-IP	TD	[101]
TCPTP	phosphatase	endo Co-IP		[102]
TPL2/COT1	kinase	Co-IP		[103]
TAK1	kinase	Co-IP		[104]
TBK1	kinase	endo Co-IP	NTD	[74]
TNIK	kinase	Co-IP	TD	[105]
TRADD	adaptor	THS, Co-IP	CTD	[89,106]
TRAF1	adaptor	THS, Co-IP	NTD, CTD	[2,89]
TRAF2	E3 ligase, adaptor	THS, Co-IP	CTD	[2,89]
TRAF3	E3 ligase, adaptor	Co-IP		[107]
TRAF4	E3 ligase, adaptor	endo Co-IP		[108]
TRIF	adaptor	THS, Co-IP		[109]
UBC13	E2	endo Co-IP	RING	[76]
USP4	DUB	Co-IP		[110]
USP7	DUB	GST		[80]
USP17	DUB	Co-IP		[107]
UXT-V1	transcriptional cofactor	endo Co-IP		[111]
VP4	rotavirus capsid protein	THS, Co-IP		[112]
WDR62	scaffold	Co-IP		[113]

Abbreviations: Co-IP, co-immunoprecipitation of transiently expressed proteins; CTD, C-TRAF domain; endo Co-IP, co-immunoprecipitation of endogenous proteins; DUB, de-ubiquitinase; GST, glutathione-S transferase pull-down assay; NTD, N-TRAF domain; RING domain; TD, TRAF domain; THS, two-hybrid system; zinc, zinc finger domain.

## Data Availability

Not applicable.

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
