# Peer review of "TNF Receptor Associated Factor 2 (TRAF2) Signaling in Cancer"

_cancers, 2022, doi:10.3390/cancers14164055_

Round 1

Reviewer 1 Report

This review article addresses an interesting and worthwhile topic.  However, as currently written it will not make a valuable contribution, for the following major reasons:

1)  The points that are made are overly repetitive, yet in a number of sections relevant prior work is not appropriately discussed.

2)  The explanations and conclusions are often provided in such a confusing manner that they are almost incomprehensible to the reader. 

3)  There are a very large number of problems with word usage, sentence structure, grammar, etc. that significantly detract from the article, and make it very hard to read/understand.

As mentioned above, some topics are discussed repetitively (e.g. TRAF2’s impact on NIK), while other relevant findings are omitted.  In general, it seems that the authors inexplicably feel the need to discuss TRAF2 as though no other TRAFs are involved in the pathways that it regulates, which leads to misleading discussions of certain points. 

Specific examples are:

1) Line 149: Another important potential explanation that should be discussed here is that TRAFs 2, 3 and 5 share an overlapping binding site on TNFR superfamily members.  Thus, overexpression of DN or WT TRAF2 interferes with the binding stoichiometry of these other TRAFs and alters the nature of the entire signaling complex. 

2) Line 153: Not entirely true.  Such redundancy was addressed in a number of earlier studies, e.g. Xie et al J Immunol 2006 and Rowland et al J Immunol 2007. 

3) Line 178: It should also be mentioned here that the same outcome results from sole depletion of TRAF3, as this TRAF is needed to recruit TRAF2 to NIK. 

4)  Paragraph ending at line 416: The authors should mention prior work showing that LMP1 signaling also requires TRAF5 for the recruitment of TRAF3; LMP1's avid binding of TRAF3 and subsequent depletion of TRAF3 for binding and inhibiting other TRAF2-binding receptors is likely an important mechanism by which LMP1 mediates its oncogenic signaling.  TRAF6 has also been shown to play a major role in LMP1 signaling. 

5) Line 430: It should be mentioned here that loss-of-function mutations of TRAF3 are in fact more common in B cell malignancies, including MM, than TRAF2 mutations. 

 6) Line 435: TRAF2 also requires support from TRAF3, for recruitment to NIK.  In fact, most of the signaling proteins that are ubiquitinated with help from TRAF2 require its association with TRAF3 for recruitment to the target. 

 7) Paragraph ending at line 559: The preceding paragraph is wandering and confusing.  Also, as discussed above, TRAF2-regulated signaling pathways in B cell malignancies would be enhanced by depletion of TRAF2, so this does not seem desirable as a systemic therapy.  These caveats should be discussed.

Author Response

This review article addresses an interesting and worthwhile topic.  However, as currently written it will not make a valuable contribution, for the following major reasons:

1)  The points that are made are overly repetitive, yet in a number of sections relevant prior work is not appropriately discussed.

2)  The explanations and conclusions are often provided in such a confusing manner that they are almost incomprehensible to the reader. 

3)  There are a very large number of problems with word usage, sentence structure, grammar, etc. that significantly detract from the article, and make it very hard to read/understand.

 As mentioned above, some topics are discussed repetitively (e.g. TRAF2’s impact on NIK), while other relevant findings are omitted.  In general, it seems that the authors inexplicably feel the need to discuss TRAF2 as though no other TRAFs are involved in the pathways that it regulates, which leads to misleading discussions of certain points. 

We fully agree with the reviewer that there is significant cooperation and/or redundancy between different TRAF proteins. We referred to such cooperation/redundancy already at several places in the first version of the manuscript (e.g. line 105; line 173; line 181; line 227, line 233, line 240, line 450, without claim for completeness). Furthermore, we ask for understanding that in a review dedicated to the role of TRAF2 in cancer, the weighting how intensively other TRAF proteins should or have to be discussed without losing the focus is also a subjective matter. Nevertheless, in our revised manuscript, we have emphasized the fact that TRAF2 could cooperate or act redundantly with other TRAFs even more and included additional references including those suggested by the reviewer (refs. 130, 131). We also included more references addressing the role of other TRAF family members in scenarios where we focused on TRAF2, especially LMP1 signaling (refs. 201-211, 213-216 – see also new text passages lines 431-445, 457-469). We furthermore reduced the cross-references to the TRAF2-NIK axis.   

Specific examples are:

1) Line 149: Another important potential explanation that should be discussed here is that TRAFs 2, 3 and 5 share an overlapping binding site on TNFR superfamily members.  Thus, overexpression of DN or WT TRAF2 interferes with the binding stoichiometry of these other TRAFs and alters the nature of the entire signaling complex.

We improved this part to:

A possible explanation for this is that other TRAF proteins, which use overlapping binding sites to TRAF2 in the considered TNFR type, act redundantly with TRAF2 and/or fulfill functions distinct of those of TRAF2.“ (lines 170-173).

2) Line 153: Not entirely true.  Such redundancy was addressed in a number of earlier studies, e.g. Xie et al J Immunol 2006 and Rowland et al J Immunol 2007.

Please note, we did not stated that redundancy/cooperation of TRAFs is not investigated at all, but only that it “has practically not addressed”. We agree that this statement is a bit exaggerated and attenuated it to “have been limitedly investigated” (line 178). We furthermore included here the examples of TRAF2/TRAF1 and TRAF2/6 cooperation/redundancy mentioned by the reviewer (lines 174-176) and the TRAF2/TRAF2 cooperation in NIK degradation (although this will be discussed in more detail later).

   3) Line 178: It should also be mentioned here that the same outcome results from sole depletion of TRAF3,

Done (lines 204/205)

as this TRAF is needed to recruit TRAF2 to NIK. 

Already explicitly mentioned above (“…TRAF2 interacts via TRAF3 with NIK…“ (line 181 and also so shown in figure 2.

4)  Paragraph ending at line 416: The authors should mention prior work showing that LMP1 signaling also requires TRAF5 for the recruitment of TRAF3; LMP1's avid binding of TRAF3 and subsequent depletion of TRAF3 for binding and inhibiting other TRAF2-binding receptors is likely an important mechanism by which LMP1 mediates its oncogenic signaling.  TRAF6 has also been shown to play a major role in LMP1 signaling. 

We discuss the roles of other TRAF proteins for LMP1 signaling in more detail in the revised manuscript (431-445, 457-469).   

5) Line 430: It should be mentioned here that loss-of-function mutations of TRAF3 are in fact more common in B cell malignancies, including MM, than TRAF2 mutations. 

Done  (lines 484 – 485)

6) Line 435: TRAF2 also requires support from TRAF3, for recruitment to NIK.

Please note, the corresponding sentence starts here with “Thus, as so often the activity of TRAF2 require support by cIAP1 and cIAP2 “. Therefore, we refer here eventually not only to the role of TRAF2 and the cIAPs in NIK degradation, where TRAF2/cIAPs cooperate with TRAF3, but also to other scenarios, where TRAF2/cIAPs do not cooperate with TRAF3 but nevertheless ubiquitinate various proteins (TNFR1 signaling). We feel therefore that this sentence is appropriate. In any case, it does not negate the role of TRAF3 in NIK degradation. Indeed, the role of TRAF2 and TRAF3 in NIK degradation has already been mentioned before.   

In fact, most of the signaling proteins that are ubiquitinated with help from TRAF2 require its association with TRAF3 for recruitment to the target. 

To our opinion, it is not substantiated by the literature to state “most signaling proteins that are ubiquitinated with help from TRAF2 require association with TRAF3”. TRAF2-mediated/dependent ubiquitination of proteins (RIPK1, TRADD, TRAF2 itself and cIAPs …) has mainly been investigated in TNFR1 signaling and there is no evidence for an involvement of TRAF3. Same applies for TNFR2-associated TRAF2-mediated ubiquitination events and TRAF2-dependent ubiquitination of caspase-8 (refs. 114, 115, 126). Thus, there are examples where TRAF2/cIAPs need TRAF3 to ubiquitinate proteins (e.g. NIK but also ULK1 or IRF5, as explicitly cited in our manuscript, refs. 116, 117) but also examples where this is not the case (refs. above). Furthermore, in many publications on TRAF2- or TRAF2/cIAP-dependent ubiquitination processes, a possible cooperative role of TRAF3 has not been investigated at all.

7) Paragraph ending at line 559: The preceding paragraph is wandering and confusing. 

We improved the readability of the conclusion paragraph.

 Also, as discussed above, TRAF2-regulated signaling pathways in B cell malignancies would be enhanced by depletion of TRAF2, so this does not seem desirable as a systemic therapy.  These caveats should be discussed.

As described in chapter 6 ”Therapeutic TRAF2 targeting” not necessarily means systemic and complete depletion or inhibition of all TRAF2-associated functions. Using agonists of TRAF2-depleting receptors, such as Fn14, would spare the B-cell compartment as Fn14 is not expressed in B-cells. Inhibitors that affect only a subset of the interactions of TRAF2 with other proteins have also the potential to leave alternative NFkappaB signaling intact in B-cells.  

 Moreover, as mentioned in our review, SMAC mimetics/IAP antagonists, which act together with TRAF2 and TRAF3 to fulfill their tumor suppressor function in B-cells, were/are in clinical studies and are well tolerated suggesting that TRAF2 (or TRAF3) inhibitors can be used clinically!  

 However, we fully agree that clinical studies with potential inhibitors of TRAF2 should spend special attention on early signs for autoimmunity or B-cell hyperplasia. We extended the conclusion paragraph correspondingly (lines 664-666).

Reviewer 2 Report

Reviewer comments:

Comments to the Author

The review article on “TNF receptor associated factor 2 (TRAF2) signaling in cancer” by Dr. Siegmund et al., summarize the various TRAF2-related signaling mechanisms and their relevance for the oncogenic and tumor suppressive activities of TRAF2. Authors have discussed the recent emerging approaches to target TRAF2 for therapeutic purposes.

• The content of the review is comprehensive and broadly discussed the TRAF2 in immune signaling pathways, its role in the control and integration of cell death programs, ER stress and autophagy, and in oncogenesis.

• The article is drafted in an appropriate manner with elaborative explanation involving appropriate references.

 Minor criticisms

• Please elaborate the abbreviations such as ICB etc.

• Please undergo a thorough check of the manuscript for typographical and grammatical errors.

Author Response

Many thanks for appreciating our work and reviewing. In the revised version we write out abbreviation when mentioned in the text first time. We also carefully checked the manuscript for typos, incorrect grammar etc.

Round 2

Reviewer 1 Report

There remain major problems with English usage and clarity in this document.  So many edits are required to make it readable and to remove/revise confusing and redundant statements that it remains difficult to accurately evaluate the scientific value of this review.